# Exogenous dsRNA-Mediated RNAi: Mechanisms, Applications, Delivery Methods and Challenges in the Induction of Viral Disease Resistance in Plants

**DOI:** 10.3390/v17010049

**Published:** 2024-12-31

**Authors:** Emmadi Venu, Akurathi Ramya, Pedapudi Lokesh Babu, Bhukya Srinivas, Sathiyaseelan Kumar, Namburi Karunakar Reddy, Yeluru Mohan Babu, Anik Majumdar, Suryakant Manik

**Affiliations:** 1Division of Plant Pathology, Indian Agricultural Research Institute, New Delhi 110012, India; lokeshbabu172@gmail.com (P.L.B.); kumarsathiya1996@gmail.com (S.K.); yelurumohanbabu@gmail.com (Y.M.B.); anikmajumdar5757@gmail.com (A.M.); suryamanik94@gmail.com (S.M.); 2Department of Plant Pathology, Junagadh Agricultural University, Junagadh 362001, India; 3Department of Plant Pathology, Professor Jayashankar Telangana State Agricultural University, Rajendranagar, Hyderabad 500030, India; bhukyasrinivas954@gmail.com; 4Department of Plant Pathology, University of Agricultural Sciences, GKVK, Bengaluru 560065, India; karunakarpath18@gmail.com

**Keywords:** exogenous dsRNA, RNA interference (RNAi), plant viral diseases, spray-induced gene silencing (SIGS), nanocarriers, dsRNA stability

## Abstract

The increasing challenges posed by plant viral diseases demand innovative and sustainable management strategies to minimize agricultural losses. Exogenous double-stranded RNA (dsRNA)-mediated RNA interference (RNAi) represents a transformative approach to combat plant viral pathogens without the need for genetic transformation. This review explores the mechanisms underlying dsRNA-induced RNAi, highlighting its ability to silence specific viral genes through small interfering RNAs (siRNAs). Key advancements in dsRNA production, including cost-effective microbial synthesis and in vitro methods, are examined alongside delivery techniques such as spray-induced gene silencing (SIGS) and nanocarrier-based systems. Strategies for enhancing dsRNA stability, including the use of nanomaterials like layered double hydroxide nanosheets and carbon dots, are discussed to address environmental degradation challenges. Practical applications of this technology against various plant viruses and its potential to ensure food security are emphasized. The review also delves into regulatory considerations, risk assessments, and the challenges associated with off-target effects and pathogen resistance. By evaluating both opportunities and limitations, this review underscores the role of exogenous dsRNA as a sustainable solution for achieving viral disease resistance in plants.

## 1. Introduction

Food production must meet a dual challenge by 2050: to increase by 60% to feed a projected global population of 10 billion while minimizing food losses caused by pests and pathogens, which currently range between 17% and 30% depending on the crop species [1]. In India alone, annual crop losses due to pests and diseases account for approximately 30% of production, translating into an economic loss of around INR 90,000 crore [2]. Viral diseases in plants significantly contribute to these losses, with global yield reductions of 10–15%, amounting to an economic impact exceeding USD 60 billion annually [1]. The emergence of new viral diseases further underscores the urgent need for innovative control strategies [3].

Recent advancements in molecular biology and genetics have yielded innovative strategies for plant protection [4]. One such groundbreaking development is the exogenous application of double-stranded RNA (dsRNA) to induce RNA interference (RNAi). This approach represents a transformative step in crop protection, offering a novel method to mitigate pathogen attacks [5,6,7,8]. RNAi, first demonstrated as a mechanism for gene silencing in *Caenorhabditis elegans* in 1998, relies on dsRNA to target specific genes [9]. Although earlier studies observed RNAi-like effects in plants, fungi, and nematodes, this discovery identified dsRNA as the fundamental trigger [10]. Subsequent research established RNAi pathways as conserved across eukaryotic organisms, playing critical roles in antiviral defense (via small interfering RNAs, or siRNAs), gene regulation (through microRNAs, or miRNAs), and genome protection against transposons (via PIWI-interacting RNAs, or piRNAs) [11].

Over the past two decades, RNAi technology has revolutionized diverse biological fields, including human medicine and agriculture [8]. In agriculture, RNAi has emerged as a promising tool for managing viral diseases, with the exogenous application of dsRNA gaining attention as an alternative to genetic transformation [5,6]. This technique offers several advantages, including greater flexibility, ease of use, and environmental safety, making it a viable solution for sustainable plant protection. Studies have consistently demonstrated the efficacy of exogenous dsRNA against plant viruses [12]. The mechanism of exogenous dsRNA-induced RNAi-based viral resistance is mentioned in Figure 1.

This review focuses on the potential of exogenous dsRNA-based RNAi technology for achieving viral disease resistance in crops. We explore the mechanisms underlying dsRNA-mediated resistance, including nucleic acid recognition and uptake in plants. Additionally, we address the production, scalability, and cost implications of dsRNA for large-scale agricultural applications. Various delivery methods, such as spray-induced gene silencing (SIGS), host-induced gene silencing (HIGS), and direct application techniques, are discussed in detail. Strategies to enhance the stability and efficacy of dsRNA are also examined, along with practical considerations for implementing this technology in viral disease management. Finally, we delve into the challenges, risk assessments, and regulatory frameworks associated with RNAi technology. By evaluating both the potential and limitations of dsRNA-based approaches, this review underscores their critical role in advancing sustainable viral disease resistance in plants and contributing to global food security.

## 2. Inducers of RNAi Pathway

Antiviral RNA silencing is a key plant defense mechanism against viruses, initiated by virus-derived double-stranded RNAs (dsRNAs) produced by RNA-dependent RNA polymerases (RDRs). These dsRNAs are processed into 21–24 nt viral small interfering RNAs (vsiRNAs) by Dicer-like enzymes (DCLs), with assistance from RNA-binding proteins like DRB4. The process is amplified by host RDRs (RDR1 and RDR6), generating secondary vsiRNAs that guide Argonaute (AGO)-containing RNA-induced silencing complexes (RISCs) to suppress viral replication. However, viruses counteract this defense by producing viral suppressors of RNA silencing (VSRs), which interfere with various steps of the pathway [13].

RNAi pathway inducers, such as mild cross-protective viral strains, hpRNAs, dsRNAs, artificial miRNAs (amiRNAs), and sRNAs, can induce viral resistance in plants through exogenous RNAi (exoRNAi), which is particularly useful in non-transgenic regions, transgenesis-recalcitrant crops, or during virus epidemics. This RNA vaccination method allows rapid screening for RNAi-efficient molecules without off-target effects, offering a novel tool to enhance plant resistance against viruses, pathogens, and pests [14].

### 2.1. Leveraging Mild Strains

Cross-protection, or mild strain cross-protection, involves inoculating plants with a mild virus variant (e.g., T36 strain variants of citrus tristeza virus, CTV) to protect against more virulent strains [15,16,17]. This method has been effectively used for managing CTV and pepino mosaic virus (PepMV) [18]. It is considered more socially acceptable than transgenic approaches but requires evaluation of durability and potential impacts on other host plants [19]. While effective against related viral isolates, rare cases of protection against phylogenetically distinct viruses have been reported [20,21].

RNA interference (RNAi) plays a key role, potentially through localized RNAi or competition for host resources [22]. Another model suggests mild strains block virion uncoating of severe strains, possibly through CP recoating (e.g., TMV) [23]. Superinfection exclusion (SIE), observed in CTV, prevents severe strains from infecting plants preimmunized with mild isolates, requiring sequence identity and suggesting nucleic acid-based mechanisms [20]. Protein-mediated SIE, linked to the CTV-encoded p33 protein, regulates viral virulence and mediates SIE [24,25]. 

### 2.2. Hairpin RNA

The discovery of RNAi has transformed crop improvement by enabling resistance against various viruses [7]. Initially, independent expression of sense and antisense RNA strands provided about 20% resistance [26]. The hpRNA strategy was developed, where linked sense and antisense strands expressed from a single promoter efficiently produce dsRNA, triggering RNAi and generating small RNAs [26,27]. Unlike earlier methods, hpRNA bypasses the need for plant-encoded RDRs for dsRNA synthesis [28].

The hpRNA transgene targeting viral genes provides strong resistance against homologous viruses [29,30] and can target multiple viruses using chimeric constructs [31]. Due to challenges in stable transformation [32], agro-infiltration is used for transient expression and screening of hpRNA constructs [33]. This method uses recombinant *Agrobacterium tumefaciens* to transiently express dsRNA, triggering systemic silencing via siRNAs [34]. These methods allow studying initiation and maintenance of PTGS, unlike transgenic approaches that primarily study PTGS maintenance [26,35,36]. Transient agro-infiltration enables comparative RNAi construct analysis without positional effects seen in stable transgenics [37], but resistance lasts only 2–4 weeks and requires repeated application. It has been successfully used to study resistance against African cassava mosaic virus and to inhibit PVY transmission by aphids [38].

### 2.3. Biogenesis and Functionality of Small Interfering RNAs (siRNAs)

Several siRNA populations, including miRNAs, phasiRNAs, tasiRNAs, and vasiRNAs, play critical roles in plant development and immunity. miRNAs, transcribed from miRNA genes, form secondary structures processed by DCL1 into mature 21–22 nt miRNAs [39,40]. These bind to AGO1-RISC to target RNA for degradation or translation inhibition [41,42]. miRNAs contribute to antiviral defense directly, such as miR168, miR395a, and miR395d in cotton, which target cotton leaf curl Burewala virus genes. Overexpression of miR398 and miR2950 suppresses symptoms of cotton leaf curl Multan virus (CLCuMuV) and its betasatellite [43,44]. In rice, miR444 indirectly enhances resistance to rice stripe virus by suppressing MADS-box proteins, activating RDR1, and amplifying RNAi signaling via secondary vsiRNA biosynthesis [45]. RDR1 also produces virus-activated siRNAs (vasiRNAs), expanding its antiviral role [39].

PhasiRNAs and tasiRNAs originate from miRNA-mediated cleavage of PHAS and TAS transcripts. These are processed by RDR6 and DCL4 into 21-nt phased siRNAs. PhasiRNAs act in cis to target source or related genes (e.g., NB-LRR), while tasiRNAs act in trans to repress unrelated genes, fine-tuning gene expression and environmental adaptation [40]. Virus entry disrupts host RNA metabolism, triggering vasiRNA production, which targets ribosomal RNA and host gene exons as an innate antiviral defense [39]. This has been observed in *Arabidopsis thaliana* with cucumber mosaic virus (CMV) 2b and turnip mosaic virus.

### 2.4. Nucleic Acid Uptake in Plants

Externally applied dsRNAs likely mimic extracellular nucleic acids from pathogens, acting as MAMPs or PAMPs to activate plant immunity via PRRs [46]. For instance, viral dsRNA [47] reduces disease symptoms by triggering PTI in *Arabidopsis* and tobacco. PTI responses include MPK activation, ROS production, and defense gene expression. Fragmented self-DNA also serves as a DAMP, eliciting defense responses [48]. *Arabidopsis* recognizes viral dsRNA through SERK1, independent of RNA silencing pathways [47]. However, the molecular mechanisms behind exogenous nucleic acid recognition, uptake, and the specific receptors involved remain unclear.

## 3. Exogenous dsRNA: Production and Cost Optimization

### 3.1. Strategies for Efficient Large-Scale Production

Efficient dsRNA or sRNA production is vital for RNAi-based crop protection. Large-scale production for biopesticides requires cost-effective methods, unlike HIGS and MIGS. Although RNAi mechanisms are absent in prokaryotes, rhizobial tRNA-derived sRNA fragments can hijack host RNAi machinery in legumes [49]. Microbial fermentation, using organisms like *E. coli* and *Pseudomonas syringae*, (Figure 2) [50] offers scalable alternatives, though challenges in yield and purity remain a concern [51,52,53]. RNAgri’s microbial technology reduces costs to $2/g compared to $60/g for in vitro synthesis using bacteriophage T7 DdRP, which is effective but expensive and suited for small-scale production [54,55,56]. Typically, 2–10 g of dsRNA per hectare is needed. Emerging technologies like bacterial minicells promise enhanced dsRNA protection and controlled release in agriculture.

### 3.2. Assessing Cost and Feasibility for Agricultural Applications

In vitro and in vivo methods using DNA-dependent RNA polymerase (DdRP) from bacteriophage T7 are widely used for dsRNA production [55]. In vitro transcription, though high-quality, is costly and limited to small-scale production, while in vivo methods utilizing microorganisms like *E. coli* and *Y. lipolytica* offer scalable alternatives [50,55]. Microbial fermentation technologies, such as those by RNAgri, enhance dsRNA stability using protective proteins, making them safer and more affordable for large-scale use.

Demand for dsRNA ranges from 2–10 g per hectare, with microbial methods reducing production costs to $2/g compared to $60/g for in vitro systems [56,57,58]. Companies like GreenLight Biosciences and Eupheria Biotech now produce large-scale dsRNA at around $1/g [59]. Innovations, including bacterial minicells, promise improved dsRNA stability and controlled release, addressing contamination and cost challenges [60,61,62].

## 4. Targeted Delivery of dsRNA

RNAi-based crop protection relies on effective dsRNA delivery, yet the mechanisms of sRNA transfer between donor and recipient organisms remain unclear. Exogenous RNAs can induce gene silencing, but the SIGS strategy faces delivery challenges. Artificial nanomaterials enhance dsRNA stability [63,64,65], although their behavior under dynamic field conditions requires further investigation [64].

### 4.1. Nanocarrier-Driven Approaches to dsRNA Delivery

Nanomaterials show promise in enhancing the efficiency of SIGS, but concerns remain regarding their biosafety, including effects on crop growth, soil biodiversity, nontarget organisms, and human health [63,66]. Studies on silver nanoparticles (AgNPs) revealed reduced seed germination and seedling growth in rice [67]. By contrast, carbon dots, with low toxicity, have shown potential for RNA delivery in plants and animals [68,69]. Carbon dots can alter the cell cycle and exhibit biotoxic effects on microbial communities. These effects include reduced enzyme activities and nitrogen removal efficiency, leading to decreased microbial diversity and abundance [63,70]. The observed impacts of nanomaterials on ecosystems highlight the need for comprehensive environmental safety evaluations for RNA biopesticides that utilize nanomaterial carriers.

### 4.2. Exploring Natural dsRNA Delivery Systems

In pharmaceutical applications, liposome-encapsulated double-stranded RNAs (dsRNAs) or small interfering RNAs (siRNAs) can safely deliver RNA cargo, although they can still trigger an innate immune response in a sequence-dependent manner, unlike naked RNAs [71,72]. To enhance RNA delivery, capsid protein homologs, such as PEG10, derived from long terminal repeat retrotransposons, have been engineered. PEG10 can bind its own mRNA, facilitating vesicular secretion, while modified PNMA2 particles, based on their capsid structure, have been engineered for efficient RNA encapsulation and delivery in mammalian cells [72,73]. Additionally, a GNA:dsRBD fusion protein, which combines the lectin Galanthus nivalis agglutinin with a dsRNA-binding domain, has been shown to improve dsRNA uptake and RNA interference (RNAi) efficacy in lepidopteran midgut cells, significantly increasing insect mortality compared to naked dsRNA [11]. Furthermore, Arabidopsis Dicer-like 3 (DCL3), known for producing 24-nucleotide small RNAs, has been found to facilitate systemic RNA silencing through its RNA-binding domain (RBD), independent of its RNA processing activity, suggesting that engineered RBD domains could be useful for dsRNA delivery [74]. These advancements highlight innovative strategies for enhancing RNA delivery and RNAi efficacy across different organisms.

### 4.3. Exogenous dsRNAs: Activating Host Immune Responses for Enhanced Defense

Exogenous dsRNAs elicit antiviral defense by activating RNAi, the primary mechanism in plants against viruses [75]. Viral replication-associated dsRNAs are degraded by DCL proteins in the RNAi pathway [75]. Additionally, dsRNAs mimic viral replicative intermediates, triggering DCL-mediated cleavage and pattern-triggered immunity (PTI). In *Arabidopsis*, in vitro-generated and virus-derived dsRNAs induce PTI by acting as pathogen-associated molecular patterns. This signaling, involving SERK1 and a specific dsRNA receptor but not DCLs, restricts viral spread through callose deposition at plasmodesmata [47]. However, viral movement proteins can suppress this host response [75]. Although dsRNAs are sequence-specific, their potential toxicity to non-target organisms should be evaluated. Additionally, the biosafety of nanomaterials used as delivery vehicles must be independently assessed to ensure environmental safety [76].

## 5. Advances in Delivery Methods for Artificial dsRNAs in Plant Systems

### 5.1. Direct dsRNA Application in Plants

#### 5.1.1. Spraying (Spray-Induced Gene Silencing—SIGS)

Spraying plants with exogenous dsRNA is one of the most promising direct application methods for RNA interference (RNAi). Exogenous dsRNA can either be taken up directly by pests or translocated from plant cells to pathogens. Locally sprayed dsRNAs have been shown to spread systemically in plants, inhibiting pathogens in untreated areas as well [54]. Enhancements such as encapsulation in hydroxide nanolayers (BioClay) increase stability against degradation and extend protection duration to over 20 days, making it more effective and environmentally friendly [77,78].

#### 5.1.2. Trunk Injection and Root Absorption

Direct injection of dsRNA into tree trunks or application to plant roots allows systemic delivery through the plant’s vascular system (xylem and phloem). These methods facilitate RNA uptake and enable targeted gene silencing, offering an alternative to spraying for certain crops [79].

#### 5.1.3. Cut Surface and Surfactant-Assisted Penetration Enhancement

Wound surfaces or cut plant tissues enhance dsRNA uptake, and surfactants like Silwet L-77 improve penetration efficiency for targeted applications [78]. Optimal dsRNA length and concentration depend on the pathogen, with viruses often requiring lengths over 200–300 nucleotides [58]. Tailored dsRNAs can target specific pathogens or related species [56]. Exogenous dsRNA applications reduce viral infection severity by altering pathogenicity [62]. Targeting viral coat protein genes delays disease onset, reduces virus titers, and provides protection for 20–70 days post-inoculation [54].

### 5.2. Symplastic and Apoplastic Delivery

The efficiency of RNAi depends on the RNA application method. Chemically synthesized 22-nt sRNAs triggered RNAi in *Nb-GFP* via high-pressure spraying but failed when delivered by petiole absorption, where sRNAs remained confined to the xylem and apoplast [80]. Similarly, a 499 nt GFP hpRNA applied through petiole absorption or trunk injection in *Vitis vinifera* and *Malus domestica* stayed restricted to these regions without initiating RNAi [80]. High-pressure spraying effectively delivers RNA for intracellular targets like plant mRNAs or viral RNAs, while petiole uptake or trunk injection retains intact dsRNAs, avoiding DCL processing, allowing pest/pathogen Dicer proteins to enhance RNAi efficacy [81,82]. This method offers a non-GMO alternative to transplastomic plants.

### 5.3. Optimizing RNA Delivery to Plants with Adjuvants

Efficient dsRNA application requires overcoming structural barriers such as stomata, cell walls, and the selectively permeable plasma membrane for internalization into plant cells. Post-endocytosis, dsRNA may be trapped in cytoplasmic vesicles, necessitating release into the cytoplasm for RNAi activation. Delivery to sufficient cells is crucial for inducing resistance, followed by movement through plasmodesmata and systemic translocation via the vascular system for maximal effect [14].

## 6. Harnessing Carrier Nanomolecules for Targeted RNA Delivery

Numerous carrier nanomolecules have been developed to overcome these barriers and facilitate dsRNA internalization. These carriers improve the stability and effectiveness of RNAi triggers by enabling cellular entry and protecting RNA from degradation.

Various nanomaterials have been explored for RNA delivery in plants, each offering unique advantages. Single-walled carbon nanotubes (SWNTs), with diameters around 1 nm, are smaller than plant cell wall pores (10–20 nm), allowing them to penetrate plant cells smoothly. The efficacy of SWNTs was demonstrated through delivery of siRNAs by binding complementary siRNA strands to nanotubes and infiltrating them into *Nicotiana benthamiana* leaves [83]. This facilitated the formation of active duplexes, leading to significant GFP knockdown, with noncovalent binding protecting the siRNAs from plant ribonucleases and enhancing silencing efficiency. Carbon dots, which are low cost and highly stable, have also shown success in drug delivery, including for plant applications. Positively charged carbon dots have been used to deliver 22 nt siRNAs, promoting efficient cellular uptake and sustained mRNA silencing in *Nicotiana benthamiana* and tomato plants [84,85]. Gold nanoclusters, conjugated with polyethyleneimine (PEI), bind siRNAs through electrostatic interactions, improving siRNA stability and promoting cellular internalization. PEI also facilitates endosomal escape via osmotic swelling from protonation, and cell-penetrating peptides further assist this process [86,87,88]. DNA-based nanostructures, including oligonucleotide tiles and 3D scaffolds, can be designed to attach siRNAs via complementary hybridization, with conjugation to SWNTs further boosting delivery efficacy [89]. Additionally, layered double hydroxide (LDH) clay nanosheets have emerged as promising carriers for dsRNA, effectively targeting viruses such as CMV and bean common mosaic virus. This formulation not only protects dsRNA from degradation but also facilitates sustained cellular uptake, offering prolonged stability under harsh conditions like UV radiation and rainfall [5].

Recent field experiments have demonstrated effective dsRNA delivery systems. PEI polymers conjugated with lipids formed nanoparticles carrying dsRNA targeting RNA Pol and CP of grapevine leafroll-associated virus-3 (GLRaV-3) [90]. Sprayed onto infected grapevines, these nanoparticles protected dsRNA from degradation, ensured systemic distribution via the phloem, reduced viral load, and improved grape quality after five biweekly applications [90]. Nanomaterials like SWNTs, carbon dots, gold nanoclusters, and LDH nanosheets enhance dsRNA stability, promote cellular uptake, and protect RNA under field conditions, enabling precise and robust RNAi-based solutions for plant viral diseases [14].

### 6.1. Spray-Induced Gene Silencing (SIGS)

Topical application of virus-specific dsRNAs has been shown to protect plants against various plant viruses [54,77,91]. This approach was first demonstrated against pepper mild mottle virus (PMMoV), alfalfa mosaic virus (AMV), and tobacco etch virus (TEV), where dsRNAs targeting viral replicase genes were mechanically applied, attenuating infections in host plants [34]. Subsequent studies have confirmed that external dsRNA application can confer viral resistance in diverse host plants against numerous viruses [58,77,91,92].

RNA-mediated virus resistance was first demonstrated [34] using in vitro transcribed dsRNA fragments targeting PMMoV replicase, TEV HcPro, and AMV RNA3 in *Nicotiana benthamiana*. Resistance required a certain dsRNA length. Applications of SIGS against different viruses along with their hosts and their targeted genes are mentioned in Table 1.

#### 6.1.1. dsRNA Formulations

The stability and efficacy of dsRNA formulations remain a significant challenge, as studies have shown that the protective antiviral effects of dsRNA often last only a few days. This short duration necessitates frequent applications, particularly for long-duration crops grown in open fields. As a result, improving the stabilization and delivery methods of dsRNA molecules has become a priority for researchers. Strategies to overcome these challenges have included virus-like particle (VLP)-based delivery, BioClay nanosheets, direct trunk injection, high-pressure spraying, and biocompatible material formulations [54,98,99].

Virus-like particles (VLPs) offer an innovative approach by encapsulating dsRNA within virus particles or virion-like structures. Companies such as Apse RNA Containers (ARCs) have utilized bacteriophage MS2 capsid proteins to produce self-assembling VLPs. These VLPs are generated in *E. coli* systems, where target RNA molecules are packaged during bacterial growth, ensuring stability and efficient delivery [54,56].

BioClay is a layered double hydroxide (LDH) nanostructure that binds dsRNA, protecting it from degradation and enabling gradual release. BioClay not only resists environmental factors like watering but also provides prolonged antiviral effects. Successful applications of BioClay include the inhibition of cucumber mosaic virus and bean common mosaic virus in *Nicotiana benthamiana* and cowpea (*Vigna unguiculata*), respectively [5].

Direct trunk injection has emerged as a delivery method particularly suitable for woody plants. Commercially available systems like Arborjet inject dsRNA formulations directly into tree trunks for systemic delivery. However, the efficacy of this method against viruses in woody plants still requires further evaluation [56,80].

High-pressure spraying is another effective strategy, allowing systemic delivery of dsRNA through plant tissues. This method induces both local and systemic silencing and facilitates secondary siRNA production, particularly when using 22 nt dsRNA molecules [58,100].

Biocompatible and environmentally friendly materials are being explored. These materials aim to enhance stability and provide a gradual release of dsRNA molecules, making them suitable for field applications [98,99].

#### 6.1.2. Challenges in dsRNA-Based Applications

The application of dsRNA-based crop protection faces challenges like short-lived antiviral effects requiring frequent reapplications, especially in open fields [58]. The high costs of in vitro synthesis, complex in vivo methods, and RNA degradation due to environmental factors like UV light and rainfall further complicate delivery [5,6,101]. Stabilization strategies such as BioClay and virus-like particles show potential but need refinement [56]. Regulatory hurdles, off-target concerns, and public perception akin to GMOs hinder adoption, highlighting the need for standardized formulations and delivery methods for consistent efficacy [55,58,98].

#### 6.1.3. Nanomaterial-Based RNAi Delivery

Nanotechnology has significantly advanced agricultural practices by introducing eco-friendly solutions for crop protection, insect and pathogen control, yield improvement, and overall crop quality enhancement. Nanoparticles, due to their stability, specificity, and unique properties, have emerged as effective tools in delivering dsRNA/RNAi for gene silencing against pests and pathogens [102].

#### 6.1.4. Nanoparticles as Nanopesticides

Nanoparticles like silver, gold, copper, zinc, and titanium are synthesized as nanopesticides to target plant-promoting disease resistance. Silica nanoparticles are widely used for manufacturing nanofertilizers and pesticides [102,103]. The insoluble active ingredient is fused with an inorganic nanoparticle coating or conjugated with polymers. Examples include fluorescent RNAi-NPs with a cationic polymer shell and polyphenylene dendrimer core [103,104,105].

#### 6.1.5. Nanocarriers for dsRNA Delivery

Nanocarriers encapsulate dsRNA to prevent enzymatic degradation and ensure specific delivery to target genes viruses. This enables post-transcriptional gene silencing for effective control [106]. Metal nanoparticles such as zinc, copper, gold, and silver disrupt the pathogen’s cell wall, induce ROS accumulation, intercalate into DNA, and deactivate proteins by binding to thiol groups [107]. Liposome-mediated gold nanoparticles, guanidine-based polymers, and detergent-formulated nanocarriers ensure nucleolytic stability, rapid cellular uptake, and RNAi penetration [108]. Nanocarriers have shown success in preventing viruses like bean yellow mosaic virus and cucumber mosaic virus [102,109,110,111].

#### 6.1.6. Types of Nanomaterials and Their Conjugates Used

Nanomaterials enhance the stability of dsRNA formulations through electrostatic interactions between positively charged nanomaterials and the phosphate groups of dsRNA. This interaction forms a nanoparticle/dsRNA complex with a net positive charge, enabling easy attachment to negatively charged cell membranes. Encapsulation by nanomaterials shields dsRNA from nuclease and chemical degradation during endocytosis [112].

#### 6.1.7. Layered Double Hydroxide (LDH) Nanosheets

LDH nanosheets, also known as BioClay or nanoclay, are aluminosilicates (e.g., bentonite, hectorite) used as nanocarriers for RNAi delivery. These nanosheets protect dsRNA from UV radiation, enhance stability and adhesion to plant surfaces, promote cellular uptake, and effectively shield plants from diseases [113,114,115].

#### 6.1.8. Boosting SIGS Effectiveness: Role of Nanoparticles in Stabilizing dsRNA

The short-term environmental stability of dsRNA is a major challenge for spray-induced gene silencing (SIGS), with research focusing on chemical modifications, delivery methods, and formulations to address this. Nanoparticles such as layered double hydroxides (LDHs) increased viral protection in tobacco from 5–7 to 20 days [114,116]. BioClay dsRNA prevented aphid-mediated bean common mosaic virus (BCMV) transmission in tobacco and cowpea, while carbon dots and chitosan improved dsRNA absorption and shelf life [5]. Artificial nanovesicles (AVs) protect dsRNA from UV radiation and nucleases, ensuring gradual release and enhanced stability, even after washing [114,117]. Though costly and complex, optimizing dsRNA formulations can improve SIGS effectiveness and scalability, making it a promising tool for crop protection [64].

### 6.2. Host-Induced Gene Silencing (HIGS)

Host-induced gene silencing (HIGS) is an RNA-based alternative to chemical pesticides and fungicides, involving transgenic expression of dsRNAs to silence target genes of pests or pathogens. HIGS has been developed to protect crops from viruses [118] and insects [119,120]. A review [121] reported effective HIGS applications, with average mortality or resistance rates of 90% for viruses and 50% for insects. Despite these advances, a major limitation is the reliance on stable plant transformation, which depends on the transformability and genetic stability of crop species [122]. Most crops lack established transformation protocols, and optimizing HIGS for new species can take years [5]. However, ongoing efforts to refine transformation methods offer promise for broader application of HIGS in crop protection [123].

#### Advances in hpRNA for Viral Disease Resistance

Dicer-like proteins (DCL3, DCL4, DCL2) process hpRNAs into small RNAs (sRNAs) for long-distance and cell-autonomous silencing [124,125,126], targeting viral coat proteins and replicase genes [127,128]. Transgenic *Nicotiana benthamiana* expressing hpRNA constructs showed resistance to citrus tristeza virus (CTV) and plum pox virus (PPV) [129,130], with field trials demonstrating efficacy against tomato yellow leaf curl virus (TYLCV) [125]. However, challenges persist, as transgenic citrus plants targeting CTV showed no resistance, likely due to host-specific factors or enhanced viral virulence [129]. Deep sequencing revealed off-target effects and transcriptome changes, emphasizing the need for optimized constructs and crop-specific promoters [124,126,131]. Applications of hpRNA against different viruses are mentioned in Table 2.

## 7. Direct dsRNA Application Techniques

RNAi is widely used in crop protection, primarily via transgenic plants, but GMO-related environmental and health concerns have limited its adoption in many countries. Non-transformative methods like spray-induced gene silencing (SIGS) provide eco-friendly, transgenic-free alternatives [91,133]. Effective gene silencing depends on delivery methods such as foliar spray, trunk injection, irrigation, seed coating, baits, or soil applications. Focus has shifted to exogenous dsRNA, siRNA, and hpRNA applications as sustainable solutions, paving the way for broader acceptance of RNAi technologies [134]. Different techniques for applying dsRNA molecules along with their applications against different viruses are mentioned in Table 3.

### 7.1. Topical Applications of dsRNA

Non-transgenic strategies use exogenous dsRNA to induce RNAi-based resistance, similar to transgenic methods [95]. For instance, dsRNA from the PRSV-Tirupati isolate targeting CP and HC-Pro genes conferred complete resistance and up to 94% protection against PRSV in papaya cv. Pusa Nanha [139]. Escherichia coli-expressed intron-containing hpRNA (CP279) also provided over 2 months of PRSV resistance in papaya [96]. Naked dsRNA degrades quickly, but nanocarriers like clay nanosheets, chitosan, and carbon dots enhance stability and systemic movement, with clay nanosheets extending protection to 20 days [5]. Delivery methods include spraying, root soaking, and nanocarrier adhesion, offering reduced off-target effects, eco-friendliness, and social acceptability, making it a promising alternative for plant disease resistance [140,141].

#### 7.1.1. Production of dsRNA for Topical Application

The synthesis of dsRNA can be achieved using in vitro or in vivo systems, employing expression vectors with essential elements like T7 or T3 DNA-dependent RNA polymerase (DdRp) promoters and multiple cloning sites (MCS) for gene incorporation. Promoters may be unidirectional (e.g., pGemT and pGemT-Easy vectors; Promega, Madison, WI, USA) or bidirectional (e.g., L4440 vector with two convergent DdRp promoters; Addgene, Watertown, MA, USA) [97].

#### 7.1.2. In Vitro dsRNA Production

The ssRNA transcription method converts sense and antisense complementary DNA (cDNA) from two plasmids into ssRNA using T7 DdRp, followed by annealing to form dsRNA [34]. The single plasmid method synthesizes dsRNA directly using *Pseudomonas syringae* dsRNA bacteriophage phi 6, which is suitable for small-scale production [101]. Different methods, their features, and kits for in vitro dsRNA synthesis are given in Table 4.

#### 7.1.3. In Vivo dsRNA Production

The in vivo method generates dsRNA within living cells by expressing inverted repeat sequences, commonly using *E. coli* strain HT115 (DE3), deficient in RNAse III and harboring an IPTG-inducible T7 DdRp for efficient dsRNA transcription [51,145,146,147]. While in vitro methods suit small-scale applications, in vivo systems are preferred for large-scale dsRNA production. Novel systems like bacteriophage-based ϕ6-expression use RdRp from ϕ6 to synthesize dsRNA from ssRNA templates, effectively managing viruses like tobacco mosaic virus (TMV) [6,101]. A modified *E. coli* system (pET28-BL21 DE3 RNase III) achieved higher yields (4.23 μg/mL) than the traditional L4440-HT115 (DE3) system (1.30 μg/mL), tripling expression efficiency [148]. And bacteriophage Phi6 RdRp enables enzymatic dsRNA production [149]. Yeast (*Saccharomyces cerevisiae*) [150], bacteria (*Corynebacterium glutamicum*) [151], and *Bacillus thuringiensis* [152,153] also offer scalable dsRNA production, which is crucial for cost-effective applications.

### 7.2. Vesicle-Mediated Cross-Kingdom RNA Trafficking

#### 7.2.1. Intercellular Communication Based on Extracellular Vesicles (EVs)

Extracellular vesicles (EVs) have gained significant attention in biological research for their role in intercellular communication, transporting waste, toxins, nutrients, and RNA for silencing [154,155]. In animal systems, EVs from mammalian cells, including adipocytes [156], cancer cells [157,158], and immune cells [159,160], transmit miRNAs and functional RNAs to regulate gene expression in target cells. These findings demonstrate the critical role of EVs in sRNA transport across animal, plant, and microbial cells, though the precise mechanisms remain unclear [161].

#### 7.2.2. RNA Components in Extracellular Vesicles (EVs)

Extracellular vesicles (EVs) transport various RNA types, including sRNAs, mRNAs, and non-coding RNAs, impacting intercellular communication [162,163]. Small RNAs, such as miRNAs and siRNAs, regulate gene expression via RNAi, participating in gene silencing and post-transcriptional regulation [164]. mRNAs carry genetic information for protein translation in recipient cells, influencing gene expression [165]. EVs also contain non-coding RNAs, like circRNAs and lncRNAs, which regulate gene expression and cellular functions through diverse mechanisms [154,166].

#### 7.2.3. Extracellular Vesicle (EV)-Mediated RNA Transport

Extracellular vesicles (EVs) are key RNA carriers in the cross-kingdom transport of small RNAs (sRNAs) between plants and pathogens, enabling targeted gene silencing [76]. The selective loading of sRNAs into EVs is mediated by specific RNA-binding proteins and signaling molecules, ensuring that only specific sRNAs are encapsulated for transport to target organisms. This selective packaging is vital for precise cross-kingdom RNA interference (RNAi) [167].

Artificial nanovesicles, designed to mimic natural extracellular vesicles (EVs), provide a customizable and efficient RNA delivery system, enhancing RNA stability and precision to strengthen plant defences against pathogens [62]. EVs protect encapsulated RNAs from extracellular RNases and enhance delivery efficiency through specific target cell recognition via surface receptors [168]. RNA degradation from rainfall, UV light, and extracellular nucleases limits spray-induced gene silencing (SIGS) effectiveness [65,169]. Nanovesicles, naturally involved in cross-kingdom RNA transport, stabilize dsRNA and allow surface modifications for targeted delivery to specific tissues or pathogens [170,171,172]. Sprayed dsRNA can be processed into siRNA or transported intact, although factors like dsRNA structure, plant physiology, and environmental conditions affect efficacy [4,173]. Nanovesicle-based stabilization mitigates these challenges by reducing dsRNA degradation, improving absorption, and enabling slow or pathogen-triggered release [5,174], making nanovesicles a promising solution for RNAi-based plant protection strategies.

EVs contain various RNAs, including miRNAs, sRNAs (18–24 nucleotides), and tyRNAs (10–17 nucleotides), indicating a sophisticated RNA transport mechanism [162]. However, the processes governing the encapsulation of unconventional non-coding RNAs, their uptake by target cells, and their role in cellular communication remain largely unexplored. Customizing RNA and vesicle composition based on plant traits and genetic factors could enhance RNAi efficacy [175]. Research on EV-mediated RNA transport is critical for improving RNA delivery efficiency and stability. Plant EVs can encapsulate endogenous molecules and exogenous therapeutic substances [90,176]. Rich in bioactive lipids, proteins, RNA, and other pharmacologically active molecules, EVs have unique morphologies and compositions that support their application in plant protection [177].

## 8. Enhancing dsRNA Stability and Effectiveness

The commercialization of dsRNA-based biopesticides faces challenges, particularly environmental instability, as dsRNA degrades rapidly due to nucleases, rain, UV rays, and microbial [64,116,136,178]. Nanoparticle delivery systems like clay nanosheets and layered double hydroxides (LDHs) enhance dsRNA stability and provide prolonged pathogen defense, as demonstrated using Cy3-labeled RNA for cucumber mosaic virus (CMV), pepper mild mottle virus (PMMoV), and bean common mosaic virus [5,97]. Cationic oligosaccharides (ODAGal4) with phosphorothioate linkages and star polycations (SPc) improve dsRNA stability and pest targeting [179,180]. Insects have been models for delivering dsRNA targeting endonucleases to prevent degradation [181]. DNA nanostructures protect siRNA from nuclease activity, and carbon dots functionalized with polyethyleneimine (PEI) and golden nanoclusters facilitate siRNA delivery via foliar infiltration [85,182]. Single-walled carbon nanotubes (SWNTs) enhance dsRNA uptake efficiency [183], while cell-penetrating peptides (CPP) maintain stability [184]. Nanomaterials like CLPs and ALPS effectively spread dsRNA in maize to mitigate viral infections [185]. UV laser-induced artificial wounds improve dsRNA mobility in tomato plant vascular systems during foliar delivery [186].

## 9. Applications of Exogenous dsRNA in Viral Disease Resistance

Exogenous dsRNA has been effectively utilized to control various plant pathogens, including viruses [6,51,187]. Studies have shown that activating RNA interference (RNAi) through the application of dsRNA, siRNA, or hpRNA can protect plants against diseases [4]. Applications of exogenous dsRNA against different viruses are mentioned in Table 5.

## 10. Risk Assessment and Regulatory Consideration

Regulatory frameworks differ across regions. In the USA, dsRNA products are classified as biochemical pesticides, requiring EPA approval under FIFRA and FFDCA [199,200]. Australia regulates them as agricultural chemical products through APVMA, while the OGTR considers SIGS non-GMO under specific conditions [201]. In the European Union, regulation follows Directive 2001/18/EC and Regulation (EC) 1829/2003 [European Parliament 2003]. HIGS-based products for food or feed fall under Regulation (EC) 1829/2003, while SIGS-based products without GMOs adhere to Regulation (EC) 1107/2009 [202]. Approval in the EU involves a two-step process: EFSA evaluates the active substance, followed by zonal assessment by Member States (MSs) [203]. Specific data requirements for dsRNA-based products are currently lacking, with evaluations based on chemical PPP standards (Regulations (EC) 284/2013 and 546/2011) [204,205]. The OECD Working Paper provides guidance on environmental risk assessments and off-target effects, but formal dsRNA-specific regulations are still pending [206]. Future adaptations may refine guidelines as understanding of dsRNA evolves.

### 10.1. GMO-free RNAi in Plants

RNAi technology holds immense potential in combating pests and pathogens in plants [207,208]. Traditionally, RNAi applications relied on recombinant viruses (virus-induced gene silencing), *Agrobacterium tumefaciens*-mediated transgenes, and transgenic plants producing dsRNA molecules for targeted host-induced gene silencing [209,210]. Notably, SmartStax Pro maize, engineered to express dsRNA against corn rootworm, was approved in 2017 by the EPA, FDA, and USDA, marking a significant milestone. However, RNAi-based transgenic crops face criticism due to anti-GMO sentiments, high commercialization costs (~USD 140 million), and limited public acceptance [211]. To address these challenges, innovative GMO-free RNAi approaches focus on the direct exogenous application of RNA molecules (dsRNA or sRNA) to plants, bypassing transgenes and recombinant viruses. These strategies aim to harness RNAi benefits while mitigating regulatory and public concerns [207,212,213,214,215,216,217]. GMO-free RNAi methods are exclusively explored in [210,211]. 

### 10.2. Safety Concerns of dsRNA Application in Plants

In plants, dsRNA can stimulate pattern-triggered immunity (PTI) independent of RNAi [218], and RNAi can trigger epigenetic changes such as RNA-directed DNA methylation (RdDM) [219]. For example, dsRNAs targeting *Arabidopsis* genes increased cytosine methylation. Long-term risk assessments are essential, as RNAi products may show delayed efficacy or non-lethal phenotypes [199,220]. Regulatory frameworks, as highlighted in [221], must address environmental fate, non-target effects, and biosafety, positioning dsRNA as a secure, eco-friendly, and targeted alternative for crop protection [222].

## 11. Challenges for Using RNAi Technology

The RNA interference (RNAi) mechanism relies on sequence homology, but siRNA can sometimes lack selectivity, leading to off-target effects, which pose challenges for plant disease management [112,223,224]. Highly conserved target genes across species increase the likelihood of off-target effects. Additionally, exogenous dsRNAs may cause epigenetic off-target effects in crops, potentially persisting transgenerationally [58]. Computational design programs and validation through bioassays are essential for assessing these effects comprehensively. Pathogens and pests can develop resistance to RNAi-based products, similar to traditional biopesticides, through mechanisms like genetic variation and single nucleotide polymorphisms (SNPs) [225]. RNAi strategies often downregulate target genes, potentially creating long-lasting resistance by limiting a pathogen’s ability to adapt. However, sequence mismatches between dsRNAs and target genes due to SNPs can reduce RNAi efficacy [226]. While synonymous SNPs incur minimal fitness penalties for pathogens, deviations in dsRNA–target compatibility diminish RNAi impact or induce resistance [113]. Hence, RNAi resistance must be carefully considered during application [227,228].

## 12. Conclusions

Exogenous dsRNA-induced RNAi offers a transformative and non-GMO approach to achieving viral resistance in plants by leveraging siRNA pathways. This technology provides precise, eco-friendly alternatives to traditional pest management practices. Efficient delivery systems, such as spray-induced gene silencing (SIGS), nanocarrier-based methods, and direct applications, have enhanced dsRNA stability and uptake, enabling practical agricultural use [5,54]. Despite its potential, challenges remain, including high production costs, environmental degradation of dsRNA, and the need for clear regulatory frameworks. Scalable production methods like microbial fermentation and nanotechnology-based stabilization reduce these hurdles, with innovations such as layered double hydroxide nanosheets and carbon dots significantly improving dsRNA durability and efficacy [56,64]. Future research should focus on optimizing delivery, ensuring safety, and addressing public concerns. Integration with other biotechnologies may expand its applications beyond viral resistance to broader plant health management. With sustained innovation, exogenous dsRNA-based technologies can revolutionize sustainable agriculture and bolster global food security [75].

## Figures and Tables

**Figure 1 viruses-17-00049-f001:**
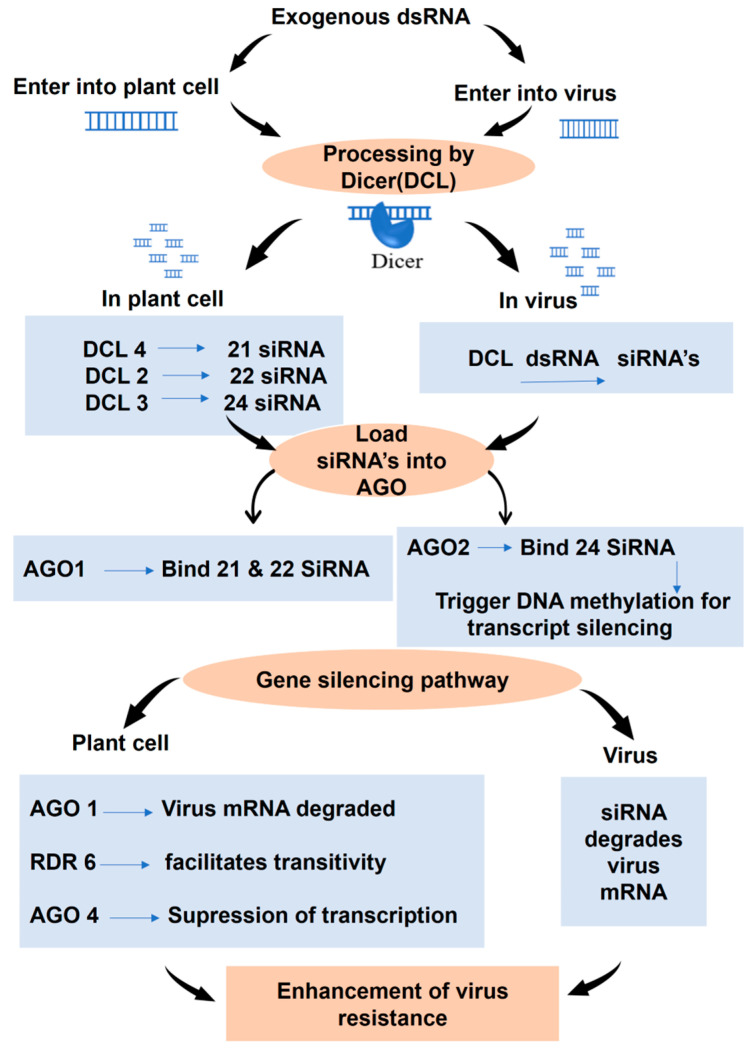
Mechanism of exogenous dsRNA-induced RNAi-based viral disease resistance in plants.

**Figure 2 viruses-17-00049-f002:**
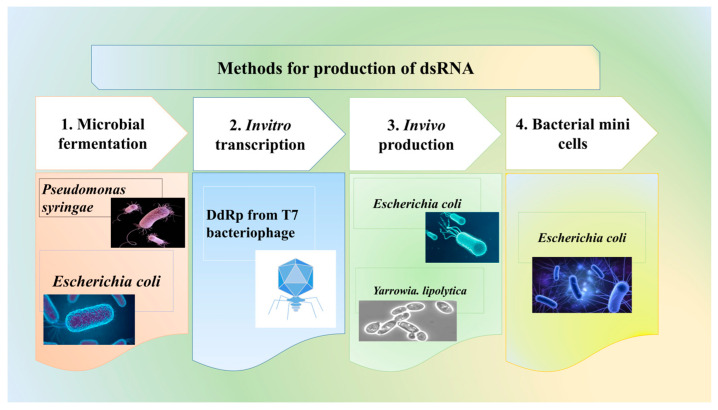
Methods for production of dsRNA.

**Table 1 viruses-17-00049-t001:** Application of SIGS against different viruses.

Plant Species	Target Virus	Targeted Gene/Region	Key Findings	Reference
Maize	Sugarcane mosaic virus (SCMV)	Coat Protein (CP)	dsRNA delivery led to virus resistance.	[51]
Pea	Pea seed-borne mosaic virus (PSBMV)	Coat Protein (CP)	Effective in reducing viral infection.	[93]
Orchid	Cymbidium mosaic virus (CymMV)	Coat Protein (CP)	Demonstrated dsRNA efficacy against CymMV.	[94]
Tobacco	Tobacco mosaic virus (TMV)	p126 Replicase	Successful virus resistance observed in tobacco plants.	[12]
Cucurbits	Zucchini yellow mosaic virus (ZYMV)	Helper Component Proteinase (HcPro)	Resistance achieved against ZYMV infection.	[95]
*Nicotiana benthamiana*	TMV	Replicase and Movement Protein (MP) (2611 bp)	Triggered antiviral responses in a dose-dependent manner.	[6]
Papaya	Papaya ringspot virus (PRSV)	Coat Protein (CP)	Effective dsRNA-based control of PRSV.	[96]
*Nicotiana benthamiana* and *Vigna unguiculata*	Bean common mosaic virus (BCMV)	-	BioClay-based dsRNA delivery provided sustained resistance to BCMV.	[97]
*Nicotiana benthamiana*	Cucumber mosaic virus (CMV)	-	BioClay nanostructures enabled gradual release of dsRNA, enhancing resistance to CMV.	[5]

**Table 2 viruses-17-00049-t002:** Applications of hpRNA against different viruses.

Target Virus	Host Plant	hpRNA Construct	Outcome	Reference
PVY	Tobacco	Intron-based hpRNA	100% silencing efficiency	[26]
TYLCV	Tomato	hpRNA targeting replicase gene	Long-lasting resistance in field	[124]
CTV	*N. benthamiana*	hpRNA targeting P23+3′UTR	Complete resistance	[129]
CTV	Citrus	hpRNA targeting P23+3′UTR	No resistance observed	[129]
PPV	*N. benthamiana*	Phloem-specific hpRNA targeting 197-bp PPV sequence	Systemic PPV resistance	[130]
TYLCV	Tomato	hpRNA targeting replicase gene	Off-target effects identified	[124]
Various RNA/DNA viruses	*N. benthamiana*	Various hpRNA constructs	Resistance in model plants but challenges in crops	[132]

**Table 3 viruses-17-00049-t003:** Different techniques for applying dsRNA molecules in plants.

Techniques	Description	Examples/Applications	References
Foliar Infiltration	Application of virus-derived small hairpin RNAs to leaves to induce RNAi and suppress pathogenicity-related genes.	Suppressed cucumber mosaic virus.	[135]
Root Absorption	Submerging roots in dsRNA suspension for uptake and translocation across the plant.	White oak seedlings showed suppressed pest incidence.	[135]
Trunk Injection	Delivery of hpRNA solution via syringe into drilled rootstocks, confirmed through microscopy.	Applied to rootstocks of *Malus domestica*.	[80]
Petiole Absorption	Uptake of small RNAs through capillary forces applied to cut stumps of petioles.	Demonstrated in *Vitis vinifera*.	[80]
Biolistic Process	Particle gun bombardment with siRNA, dsRNA, or DNA encoding hpRNA to trigger RNAi resistance pathways against pathogens.	Protected pea plants against pea seed-borne mosaic virus; reduced virus concentration significantly.	[136,137]
Agroinoculation	Injection of *Agrobacterium tumefaciens* with RNA constructs into intracellular leaf spaces to induce RNA silencing.	RNA silencing in various plants.	[138]
Seed Soaking	Soaking seeds in dsRNA solution to alter plant growth and gene expression.	Targeting CTR4sv3 in tomatoes altered growth and gene expression.	[134]
Fruit Injection	Injection of dsRNA into green tomato fruits for non-transgenic manipulation of specific pathways, such as the ethylene pathway.	Effective ethylene pathway manipulation in tomato plants.	[134]

**Table 4 viruses-17-00049-t004:** Different methods, features, and kits for in vitro dsRNA synthesis.

Method/System	Key Features	Examples/Kits	References
In vitro enzymatic synthesis	Utilizes complementary single-stranded RNA (ssRNA) synthesized and annealed to form dsRNA.	MEGAscript^®^ RNAi Kit (Life Technologies) Replicator™ RNAi Kit (Finnzymes) T7 RiboMAX™ Express (Promega, USA)	[34,142]
T7 DNA-dependent RNA polymerase (DdRP)	Transcribes target sequences from cDNA templates extracted from virus-infected plants.	Requires specific primers with T7 promoter or plasmids carrying viral sequences.	[12]
Dicer-like (DCL) enzyme digestion	Cuts long dsRNA into small interfering RNAs (siRNAs) of 18–27 nt length.	ShortCut^®^ RNase III (NEB) PowerCut Dicer (Thermo Scientific)	[45,143]
Highly processive RNA-dependent RNA polymerase (RdRP)	Coupled with T7 polymerase to allow de novo primer-independent initiation of ssRNA synthesis.	Utilizes RdRP from bacteriophage ϕ6 for large-scale dsRNA production.	[101,144]
Optional siRNA purification	siRNAs can be cleaned using kits for further applications.	mirVana™ miRNA Isolation Kit (Life Technologies)	[45]

**Table 5 viruses-17-00049-t005:** Exogenous application of dsRNA against different viruses.

S.No	Pathogen	Target Genes	Host	Reference
1	Tomato yellow leaf curl virus (TYLCV)	Topical application of specific dsRNA significantly reduced disease incidence.	Tomato	[188]
2	Papaya ringspot virus (PRSV)	dsRNA targeting PRSV genes provided 100% resistance to PRSV-Tirupati and 94% resistance to PRSV-Delhi.	Papaya	[139]
3	Tobacco mosaic virus (TMV)	dsRNA molecules derived from TMV p126 and coat protein genes conferred resistance.	Tobacco	[12]
4	Cucumber mosaic virus (CMV)	dsRNA targeting 2b gene and conferred resistance.	Pepper	[189]
5	Citrus tristeza virus (CTV)	dsRNA targeted CP p20 and p23 genes and conferred resistance.	Sweetorange	[189]
6	Potato virus X (PVX)	dsRNA targeted CP gene and conferred resistance.	Tobacco	[190]
7	Tomato Leaf Curl New Delhi Virus (ToLCNDV)	dsRNA targeted N gene and conferred resistance.	Zucchini	[191]
8	Mungbean yellow mosaic virus (MYMV)	dsRNA targeted CP, Rep genes and conferred resistance.	Blackgram	[192]
9	Pigeonpea sterility mosaic virus (PPSMV)	dsRNA targeted RdRp (606 bp), NP and MP genes and conferred resistance.	Pigeonpea	[193]
10	Sesbania mosaic virus (SeMV)	dsRNA targeted CP, MP genes and conferred resistance.	Sesbania	[194]
11	Tomato yellow leaf curl virus (TYLCV)	dsRNA targeted V2, C4 genes and conferred resistance.	Tomato	[188]
12	Cucumber mosaic virus (CMV)	dsRNA targeted CP, 2b genes and conferred resistance.	Tobacco	[195]
13	Papaya ring spot virus (PRSV)	dsRNA targeted HC-Pro, CP genes and conferred resistance.	Papaya	[139]
14	Tomato mosaic virus (ToMV)	dsRNA targeted CP gene and conferred resistance.	Tomato	[196]
15	Tomato spotted wilt virus (TSWV)	dsRNA targeted N gene and conferred resistance.	Tobacco	[197]
16	Bean common mosaic virus (BCMV)	dsRNA targeted Nib, CP genes and conferred resistance.	Tobacco	[97]
17	Tomato leaf curl virus (TLCV)	dsRNA targeted AC1/AC4, AV1/AV2 genes and conferred resistance.	Tomato	[198]
18	Zucchini yellow mosaic virus (ZYMV)	dsRNA targeted HC-Pro, CP genes and conferred resistance.	Cucumber	[95]

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
