# Peer review of "Exogenous dsRNA-Mediated RNAi: Mechanisms, Applications, Delivery Methods and Challenges in the Induction of Viral Disease Resistance in Plants"

_viruses, 2024, doi:10.3390/v17010049_

Round 1
Reviewer 1 Report
Comments and Suggestions for Authors
This review article comprehensively covers virus suppression techniques utilizing gene silencing, making it an informative and valuable resource. However, there are numerous instances where the formatting does not comply with the required guidelines, and revisions are necessary. For example, the citations are formatted with brackets [ ] instead of parentheses ( ). The text in Fig. 1 appears compressed a little, and the tables are difficult to read due to excessive line breaks. Therefore, revisions to the figures and tables are necessary to improve its quality.
Minor points
The format of the citations should be reviewed. Brackets were used instead of parentheses, and some citations appeared after periods.
‘in vitro’, ‘in vivo’ should be italic.
L.147 CMV → cucumber mosaic virus (CMV)
L.179 (RNAgri, 2020) → the citation is wrong.
Reviewer 2 Report
Comments and Suggestions for Authors
This is a good quality review paper about exogenous dsRNA mediated RNAi for agriculture application. Here, I would like to give some advice to further improve the manuscript:
1, the Figure is too simple and not very aesthetically pleasing, suggest redrawing it.
2, a brief introduction about the RNA silencing can be introduced before or at the begin of section 2 (inducers of RNAi pathway).
3, there are too many small sections, e.g., 4.2.1-4.2.4, which can be combined. Also, inside section 6, there are also five points, I will suggest to combine these into one paragraph to increase readability.
4, one or two additional figures about methods for production of dsRNA or application of dsRNA can be added to the paper.
Reviewer 3 Report
Comments and Suggestions for Authors
See attached a word file.

Needs English improvements.
